# *Cordyceps militaris*: An Overview of Its Chemical Constituents in Relation to Biological Activity

**DOI:** 10.3390/foods10112634

**Published:** 2021-10-30

**Authors:** Karol Jerzy Jędrejko, Jan Lazur, Bożena Muszyńska

**Affiliations:** Department of Pharmaceutical Botany, Faculty of Pharmacy, Jagiellonian University Medical College, 9 Medyczna Str., 30–688 Kraków, Poland; karol.jedrejko@gmail.com (K.J.J.); janlazur@gmail.com (J.L.)

**Keywords:** *Cordyceps militaris*, bioactive compounds, biological activities, cordycepin, ergothioneine, polysaccharides

## Abstract

Cordyceps spp. mushrooms have a long tradition of use as a natural raw material in Asian ethnomedicine because of their adaptogenic, tonic effects and their ability to reduce fatigue and stimulate the immune system in humans. This review aims to present the chemical composition and medicinal properties of *Cordyceps* *militaris* fruiting bodies and mycelium, as well as mycelium from in vitro cultures. The analytical results of the composition of *C*. *militaris* grown in culture media show the bioactive components such as cordycepin, polysaccharides, γ-aminobutyric acid (GABA), ergothioneine and others described in the review. To summarize, based on the presence of several bioactive compounds that contribute to biological activity, *C.* *militaris* mushrooms definitely deserve to be considered as functional foods and also have great potential for medicinal use. Recent scientific reports indicate the potential of cordycepin in antiviral activity, particularly against COVID-19.

## 1. Introduction

*Cordyceps* spp. mushrooms, classified into the *Ascomycota* group, have a long tradition of use as a natural agent in Asian ethnomedicine because of their adaptogenic and tonic effects and their ability to reduce fatigue and stimulate the immune system in humans. Among the numerous, approximately 500, species of *Cordyceps*, most scientific research has been conducted on *Cordyceps sinensis*, which is included in the Chinese Pharmacopoeia (2015), and its adenosine concentration is considered as the main quality indicator. Considering several complications (explained further in this review) regarding *C. sinensis*, the species *C. militaris* has attracted more attention from scientists and the industry over recent years [1,2].

*C. sinensis* is an authorized ingredient in food and dietary supplements in the European Union (EU), and it is included in European Commission (EC) and European Food Safety Authority (EFSA) documents such as the Novel Foods Catalogue, Compendium of Botanicals. Health claims for *C. sinensis* have been proposed in the past, such as that *C. sinensis* possesses antioxidant properties (400–800 mg/day), stimulates the immune system and increases performance and endurance during exercise (3 g dried powder or equivalent extracts); however, these claims have not been approved by EFSA and EC, because the cause-and-effect relationship between the consumption of *C. sinensis* and the corresponding health benefit has not yet been established. Meanwhile, *C. militaris* is not included in the abovementioned documents in the EU, meaning it is considered an unauthorized ingredient of foods or dietary supplements [3].

The inhabitants of China, Tibet, Nepal and India have consumed *Cordyceps* spp. for centuries in order to adapt their bodies to difficult high mountain conditions such as low ambient temperature, high atmospheric pressure and reduced oxygen content in the environment. Traditional Chinese medicine recommends the use of *Cordyceps* spp. for treating several human disorders such as cardiovascular and respiratory diseases, disorders of the liver and kidney, cancers, diabetes, infectious and parasitic diseases and sexual dysfunctions [1,2].

A unique aspect of *Cordyceps* spp. is its existence as an entomopathogenic fungus in the natural environment. These species are endemic to the Himalayas at locations 2000 m above sea level. *C. sinensis* is a parasitic fungus that attacks the larvae of moths *Thitarodes* spp. and *Lepidoptera* spp. These larvae reside approximately 15 cm underground, where they become “infected” with *C. sinensis*, which gradually uses the entire interior part of the larvae and then produces fruiting bodies on their shells so that these are visible above the soil surface. The indigenous people collect *C. sinensis* with the larva in the season from May to June, dry it, pulverize it and then prepare an extract with hot water. After that, the preparation can be consumed as a hot water beverage [2].

Because of the limited resources of *C. sinensis*, endemic occurrence, unique growth pattern and biology of its life cycle in the ecosystem, the long-term process of symbiosis between the fungus and the larva (from autumn to spring) is time-consuming, and the harvesting process of this species from its natural site results in a very high price. Obtaining this species from the natural environment has become insufficient to meet the growing demand for raw materials for the production of dietary supplements, nutraceuticals and functional foods, especially in Asia, the United States and Europe [2,4,5].

Since historical times, *C. sinensis* has been used as an ingredient of food and medicine and is therefore more recognizable by consumers. The aforementioned difficulties in obtaining *C. sinensis* from culture media, the growing demand for this species and its high prices due to limited natural resources have led to the search for alternative sources. The solution to this issue is a related species, namely *C. militaris* (L.) Fr., which can be grown and obtained in vitro. Importantly, it was observed that under in vitro culture conditions, *C. sinensis* mycelium cannot synthesize cordycepin (or can synthesize only in a negligible amount) and has a lower content of amino acids and D-mannitol than those in fruiting bodies of this species obtained from natural sites [6].

*C. militaris* is an alternative to *C. sinensis* because the qualitative and quantitative composition of bioactive substances from in vitro-cultivated *C. militaris* does not differ from the content of these substances in *C. sinensis* fruiting bodies. Therefore, the aim of this work was to review the composition of *C. militaris*. The analysis of the composition of *C. militaris* from the culture media showed that the concentration of cordycepin and polysaccharides is higher than that in *C. sinensis* from the natural environment [7,8]. Both species show a similar profile of 17 amino acids, with the predominance of L-arginine and L-proline in *C. militaris* from the culture media [2,5].

## 2. Materials and Methods

Keywords used for data collection were: *Cordyceps militaris*, bioactive compounds, biological activities and cordycepin. The following databases were examined: US National Library of Medicine (PubMed), Medical Literature Analysis and Retrieval System Online (MEDLINE), EMBASE, Scopus and Google Scholar. The search terms were applied as keywords in the titles, abstracts and body of the journal articles in academic databases, including Cochrane Library, Frontiers, Jagiellonian University Repository, PLoS ONE, MDPI, Wiley Online Library, SAGE Journals, ScienceDirect, Springer Link and Taylor and Francis. The selection criteria for studies to be considered in the review include that the study is written in English (unless, for relevance to the review, it was available in a different language), the study is published in a peer-reviewed journal and the study is published between 1991 and 2021 (unless its historical value and, therefore, older scientific works were also used).

## 3. Bioactive Compounds from *C. militaris*

Based on the results of the scientific research, it is possible to obtain information on bioactive compounds from the group of nucleosides and polysaccharides contained in *C. militaris*. Nucleosides—cordycepin and adenosine—have been confirmed to be present in *C. militaris*, and the content of these ingredients in *C. militaris* is higher than those in *C. sinensis*. The analytical results showed the presence of the following biologically active substances in *C. militaris*: γ-aminobutyric acid (GABA) and ergothioneine; glycolipids (cerebrosides), glycoproteins (lectins), D-mannitol (referred as cordycepic acid), xanthophylls including carotenoids (lutein and zeaxanthin), sterols (ergosterol), statins (lovastatin), phenolic compounds (including phenolic acids and flavonoids), vitamins and biominerals/bioelements (magnesium, potassium, selenium, and sulfur) [7,8,9].

Previous studies have shown that the concentration and distribution of bioactive compounds is not uniform in the fruiting bodies. The outer parts of *C. militaris* fruiting bodies have the highest concentration of nucleosides, polysaccharides, carotenoids and selenium organic compounds. The comparison of the content of bioactive ingredients in the mycelium and fruiting bodies of *C. militaris* is presented in Table 1. The optimal drying temperature for *C. militaris* is 60 °C. A higher temperature causes a loss of the content of cordycepin and phenolic compounds [10].

### 3.1. Nucleosides

Cordycepin (3ʹ-deoxyadenosine) is a water-insoluble organic compound and a structural analog of the nucleoside adenosine (Figure 1).

Cordycepin was isolated from *C. militaris* in 1950. On the basis of in vitro and in vivo studies performed to date, this compound has been confirmed to exhibit the following activities: immunostimulating, anti-inflammatory, antiviral, antitumor, ergogenic, hypolipidemic, hypoglycemic and regulation of steroidogenesis and spermatogenesis. Few studies have demonstrated the antioxidant activity of cordycepin. The antioxidant activity has been explained in scientific studies based on the mechanisms of biological activity of polysaccharide fractions contained in fruiting bodies of *C. militaris*. Experiments conducted on rodents demonstrated that the immunostimulatory activity of cordycepin results from its ability to induce cellular and humoral immune response. The studies revealed an increase in the concentration of interleukins IL-4, IL-10 and IL-12, and Th1 and Th2 cytokines, a decrease in the concentration of IL-2 and transforming growth factor-β (TGF-β) and an increase in the level of T lymphocytes (CD4 and CD8). One of the mechanisms of action of cordycepin is supplementing “produced” energy in the form of adenosine-5′-triphosphate (ATP) molecules. Cordycepin can also increase the concentration of nitric oxide (NO) [11,12].

### 3.2. Carbohydrates

D-Mannitol is one of the most important products of *C. militaris* metabolism and is classified as a polyhydric alcohol (polyol). D-Mannitol from *C. militaris* is commonly referred to as cordycepic acid. It is used by *C. militaris* as a carbohydrate reserve and as a transporter of other compounds in the osmoregulation and control of metabolic pathways. D-mannitol shows osmotic activity, thus it can be used in clinical practice as a diuretic and anti-edematous drug [7,13].

In addition to D-mannitol, *C. militaris* contains saccharides with more complex structures—polysaccharides. Depending on the site of polysaccharide biosynthesis in the hyphae of *C. militaris*, these may occur as secondary intracellular (IPS) or extracellular (EPS) metabolites. Polysaccharides isolated from the mycelium of *C. militaris* have a different chemical structure. The chemical structure is determined by the type of monosaccharides that form the polysaccharides, their linear sequence, spatial configuration, location of glycosidic bonds and degree of branching (cross-linking) of the chain [14].

Significant differences have been observed in the chemical structure as well as the qualitative and quantitative composition of monosaccharides obtained from the fraction of polysaccharides present in *C. militaris* growing in the natural environment, and cultivated under laboratory conditions. Mannose, glucose and galactose are the most important monosaccharides which form saccharide polymers. In subsequent scientific studies on the polysaccharides of *C. militaris*, the following other monosaccharides were detected: arabinose, rhamnose and xylose [14].

In the CPSN Fr II polysaccharide obtained from the culture broth of *C. militaris*, the percentage content of monosaccharides is as follows: mannose (65.12%), galactose (28.72%) and glucose (6.12%) [15]. The WCBP50 polysaccharide structurally contains α-D-glucose, α-D-mannose and α-D-arabinose units linked by an α-glycosidic bond [16]. CMN1 is a 37.8 kDa polysaccharide isolated from *C. militaris* mycelium. CMN1 is composed of the subunits of D-galactose, D-mannose, L-arabinose and L-rhamnose linked by a glycosidic bond (1 → 2) and (1 → 3). The branching of the side substituents is in the (1 → 4) and (1 → 6) positions [17].

The basis of the structure of LCMPs-II is (1 → 4)-α-D-glucose with side substituents (1 → 3) of rhamnose, xylose and glucose [18]. In subsequent studies, the SeCSP-I polysaccharide fortified with selenium was obtained, and the structure of this polysaccharide was shown as (1 → 4) galactose, (1 → 3) and (1 → 6) mannose and (1 → 4) glucose [19]. Subsequent studies showed that the CMP-W1 polysaccharide consists of 55.4% D-mannose, 19.5% D-glucose and 25.1% D-galactose. The types of glycosidic bonds identified in the chain are (1 → 3), (1 → 4) and (1 → 6) glycosidic bonds [20].

The CMP-1 polysaccharide has a low molecular weight of 4.4 kDa and consists of the following combined units: (1 → 4) α-D-glucose, (1 → 6) β-D-glucose and (1 → 4) β-D-glucose, and the side branches contain (1 → 3) α-L-rhamnose [21]. The new structure CMPB90-1 isolated from *C. militaris* was found to be composed of α-D-glucose linked by (1 → 6) and (1 → 3) side branches containing (1 → 4) β-D-mannose [22].

To summarize, the differences in the obtained results for the chemical structure of polysaccharides are due to the source of *C. militaris*, the cultivation methods and conditions and the extraction method. The molecular weight, spatial conformation, type of glycosidic bonds and the degree of branching (cross-linking) of polysaccharides influence the biological activity of *C. militaris* polysaccharides [14].

In vitro and/or in vivo experiments showed that the polysaccharides contained in *C. militaris* exhibit immunostimulatory, antitumor, anti-inflammatory, antioxidant, hypoglycemic, hypolipidemic, and hepatoprotective activities. These studies proved the immunostimulatory activity of the polysaccharide fractions isolated from *C. militaris*, and the tested polysaccharides were shown to induce an immune response in vitro after stimulating the activity of macrophages to produce NO, IL-1β, interferon (IFN -γ), tumor necrosis factor (TNF-α), T and B lymphocytes and natural killer (NK) cells as well as increased phagocytosis by macrophages [14].

The experiments confirmed that the polysaccharides exerted antitumor activity through growth inhibition and induction of apoptosis of tumor cells. Inhibition of carcinogenesis involved inhibition of cyclin-dependent kinase and arrest of the tumor cell cycle in the G0/G1 and G2/M phases [14].

### 3.3. Amino Acids

The total content of amino acids in fruiting bodies of *C. militaris* is 57.39 mg/g dry weight. In addition to protein amino acids, the fruiting bodies contain non-protein amino acids, such as GABA and ergothioneine [9].

GABA is a non-protein amino acid biosynthesized in the human body from glutamic acid (glutamate). GABA is an inhibitory neurotransmitter in the central nervous system and affects various parts of the nervous system, including the cerebellum, hippocampus, hypothalamus, striatum and spinal cord. GABA interferes with the activity of GABAergic receptors subtype A, B and C. It regulates sleep, memory and learning processes and emotional processes such as anxiety and stress. GABA also shows myorelaxant and anticonvulsant activities [23].

Previous scientific research has shown discrepancies in the results on the pharmacokinetic and pharmacodynamic parameters of GABA after oral administration in humans. Some studies indicate the low bioavailability of GABA after oral administration. Other complications related to GABA include difficulty in penetrating the blood–brain barrier and short biological half-life [24].

Some scientific publications have confirmed the bioavailability and effectiveness of oral GABA supplementation in humans [25,26].

In studies on mycelium and fruiting bodies of *C. militaris*, the concentration of GABA was determined, respectively, at 68.6–180.1 mg/kg and 756.30 μg/g dry weight [7,8].

Ergothioneine (2-thiol-L-histidine-betaine) is a water-soluble sulfur analog of the amino acid L-histidine with an attached fragment of the betaine molecule (Figure 2).

Ergothioneine is a non-protein amino acid produced by certain bacteria, plants, and fungi, but it is not produced by mammals and therefore must be provided through the diet [27].

A valuable source of ergothioneine are various species of fungi, in which the concentration of this substance ranges from 0.2 mg/g to 2.6 mg/g dry weight. The content of ergothioneine in *Agaricus bisporus* is approximately 0.55 mg/g dry weight. The concentration of ergothioneine has been confirmed in the following species: *Lentinula edodes* (shiitake), 1.98 mg/g dry weight; *Pleurotus osteratus* (oyster), 2.59 mg/g dry weight, and *Grifola frondosa* (maitake), 1.13 mg/g dry weight [27,28,29].

The concentration of ergothioneine in the fruiting bodies of *C. militaris* was determined to be 782.3 mg/kg dry weight, whereas in mycelium it achieves range 130.6 mg/kg dry weight [9]. In another study, it was estimated to be 409.8 μg/g dry weight in the fruiting bodies of *C. militaris* [8].

The human body is unable to biosynthesize ergothioneine. A specific transporter carnitine/organic cation transporter 1 (OCTN1) for this substance has been identified, and a high concentration of ergothioneine has been confirmed in some tissues and cells, including erythrocytes, spleen, liver, and eyes. The antioxidant, cytoprotective, and radioprotective activities of ergothioneine have been demonstrated in in vitro and in vivo experiments on animal models [30].

In healthy volunteers aged 21–35 years, supplementation with ergothioneine (at the dose of 5 mg/day and 25 mg/day) was found to correlate with a slight reduction in the level of oxidative stress markers. Presumably, the antioxidant activity of ergothioneine may be more important in situations predisposing to the generation of oxygen free radicals, e.g., diseases with inflammatory processes or physical exercise [31].

The concentration of ergothioneine in the human body decreases with age [32]. Ergothioneine concentration significantly decreases in the elderly people after 60 years of age, which may correlate with the development of neurodegenerative diseases [33].

### 3.4. Carotenoids

The presence of carotenoids, including xanthophyll derivatives, has been confirmed in fruiting bodies of various species of fungi, including *C. militaris*. Carotenoids are responsible for an intense yellow-orange color of *C. militaris* fruiting bodies. The main xanthophylls present in the fruiting bodies of *C. militaris* are β-carotene, lycopene, lutein and zeaxanthin [34,35]. Lutein and zeaxanthin are also found in the macula of the human eye, which is a cluster of cones responsible for color (daytime) vision [36]. In addition to research on the protective effect of lutein and zeaxanthin on eye structures, previous scientific reports have indicated that supplementation with carotenoid pigments improves cognitive functions, reduces the level of cortisol and stress symptoms in young and adult people, and induces antioxidant effects [36,37,38].

It has been proved that a diet rich in carotenoids, such as the consumption of tomato juice (high concentration of lycopene) and carrot juice (high concentration of β-carotene), correlates with a positive effect on immune functions in healthy males who previously showed a low level of carotenoids in their diet [39]. There are discrepancies in the effects of β-carotene on human health. Some reviews indicate a benefit of β-carotene supplementation in the area of cancer diseases [40], whereas in some scientific reviews contradictory information can be found in that supplementation of β-carotene does not have a significant effect in the prevention of cancer, and conversely, in some types of cancer, such as lung or stomach cancer, it may increase the risk of cancer. It should be noted that the increased risk of lung or stomach cancer was found in the case of supplementation with high doses of β-carotene, in the range of 20–30 mg per day [41].

It was confirmed that lycopene supplementation has a positive effect on the functions of the vascular endothelium in patients with cardiovascular disease (CVD). However, no significant impact on endothelium function in healthy volunteers [42]. The correlation between the supply of lycopene and the prevention of the metabolic syndrome was confirmed [43]. In vitro and in vivo studies have shown that lycopene inhibits the progression and induces apoptosis of prostate cancer cells. Lycopene is an auxiliary agent supporting basic chemotherapy and hormone therapy in patients with prostate cancer [44].

The concentration of β-carotene and lycopene in the extract of fruiting bodies of *C. militaris* was determined. The content of β-carotene and lycopene was 0.328 mg/g and 0.277 mg/g, respectively [45]. Choi et al. [46] noted that in the aqueous extract of *C. militaris*, the concentration of β-carotene was determined at 24.51 μg/g, whereas lycopene was 3.42 μg/g.

For comparison, the concentration of β-carotene in *A. bisporus* (Turkey) is 0.04 mg/g, whereas in *Pleurotus ostreatus* (India) it is 0.03 mg/g [35]. Content of β-carotene has been demonstrated in the species *Tricholoma acerbum* at 75.48 μg/g, whereas lycopene is at 39.65 μg/g [47].

In fruiting bodies of *C. militaris*, the presence a novel class of carotenoids was confirmed such as cordyxanthins. Four cordyxanthins have been identified in fruiting bodies of *C. militaris*, which are marked numerically as cordyxanthin I–IV. Concentration of cordyxanthin I, cordyxanthin II, cordyxanthin III and cordyxanthin IV was, respectively, 0.289 mg/g, 0.235 mg/g, 0.401 mg/g, 0.175 mg/g. Unlike lutein, zeaxanthin, β-carotene and lycopene, cordyxanthins show better solubility in water, which results from differences in the chemical structure—they contain less lipophilic methyl groups and more hydroxyl substituents. Among the marked cordyxanthins, cortixanthin II deserves attention because unlike most carotenoids, it does not present in two, six-membered carbon rings, but one six-membered and the second five-membered carbon structure [34].

### 3.5. Statins

The fruiting bodies of *C. militaris* are a valuable source of lovastatin that belongs to a group of compounds called statins, which are commonly used as cholesterol-lowering drugs [7,8]. Lovastatin is a naturally occurring compound that selectively blocks the synthesis of endogenous cholesterol. It was first isolated from *Aspergillus terreus* in 1978 and introduced as a medicinal compound by Merck in 1987. Lovastatin contains a six-part lactone ring with a hydroxyl group and partially hydrogenated naphthalene with a hydroxyl substituent esterified with a 2-methylbutyric acid residue (Figure 3). Lovastatin is a precursor converted from a lactone form to a hydroxy acid during enzymatic reactions, which competitively blocks 3-hydroxymethylglutaryl-coenzyme A (HMG-CoA). Consequently, the conversion of HMG-CoA to mevalonate, which is a key step in cholesterol synthesis, is blocked. Drugs from the statin group also show pleiotropic effects, which include protection of the vascular endothelium. Lovastatin is an approved drug at the dose of 20 mg and is indicated in the primary treatment of hypercholesterolemia in patients and in reducing the development of coronary atherosclerosis in patients with coronary artery disease [48,49].

The concentration of lovastatin in mycelium of *C. militaris* is defined in the range from 37.7 mg/kg to 57.3 mg/kg dry weight. In this respect, *C. militaris* has a lower content of lovastatin than that determined in *C. sinensis* at 1365.3 mg/kg [8]. For comparison, in fruiting bodies of *C. militaris* the concentration of lovastatin is 2.76 μg/g. However, one of the highest contents of lovastatin was confirmed in fruiting bodies of *Hericium erinaceus* 14.38 μg/g and *Ganoderma lucidum* 11.54 μg/g [7].

### 3.6. Phenolic Compounds

An important group of phenolic compounds present in mushrooms are phenolic acids and flavonoids. Their potential is associated with a strong antioxidant effect and the ability to protect important structures such as proteins, enzymes, lipids or nucleic acids against oxidative damage. The strongest antioxidant properties are exhibited by phenolic acids such as vanillic and caffeic acids. p-Hydroxybenzoic, gallic and protocatechuic acids found in edible mushrooms not only show antioxidant activity but also exert antibacterial, antifungal, antiviral and anti-inflammatory effects as demonstrated by in vitro and in vivo studies [50].

The total content of phenolic acids and organic acids in *A. bisporus* is 23.43 µg/g, with the concentration of gallic acid reaching only 0.83 µg/g. The total amount of phenolic acids and organic acids in *P. ostreatus* was determined to be 10.41 µg/g. The concentration of gallic acid and protocatechuic acid was determined at 1.38 µg/g and 2.07 µg/g, respectively [51]. The concentration of phenolic acids in fruiting bodies of *C. militaris* was also analyzed. The concentration of p-hydroxybenzoic acid is 0.02 mg/100 g dry weight and cinnamic acid 0.11 mg/100g dry weight were determined [52].

The concentration of flavonoids was determined in the extract of fruiting bodies of *C. militaris* in the amount of 1.56 mg/g [45]. The concentration of flavonoids was calculated as rutin equivalent (RE). The content of flavonoids in fruiting bodies and mycelium was determined, respectively, as 5.54 mg RE/g and 2.26 mg RE/g [53]. In the aqueous extract of *C. militaris*, the concentration of flavonoids was 275.52 mg/g, whereas polyphenols were 19.79 mg/g [46]. The concentration of flavonoids is higher in the extract than in fresh fruiting bodies of *C. militaris*. The total content of flavonoids was 6.6 mg RE/100 g and 4.5 mg RE/100 g respectively in the extract and fresh fruiting bodies of *C. militaris* [54].

### 3.7. Other Bioactive Compounds in C. militaris Fruiting Bodies

Lectins, detected in fruiting bodies of *C. militaris* on the basis of their chemical structure, are protein molecules with attached saccharide fragments (glycoproteins). Lectins have been demonstrated to possess mitogenic activity. These bind to sugar residues on the surface of cells to initiate the process of cell clumping—the agglutination of cells [55,56]. Another group of compounds found in this species are beauveriolides, characterized by a complex chemical structure—cyclodepsipeptides. Beauveriolides exhibited antiatherosclerotic activity and the ability to reduce β-amyloid concentration [57]. Militarinones are a less known group of chemical compounds classified as alkaloids. Structurally, these are pyridine derivatives or tetramic acids (pyrrolidine-2,4-dione) and have been shown to possess antimicrobial and cytotoxic activities [57].

Pentostatin isolated from the fruiting bodies of the described species is an analog of the purine base hypoxanthine. It exhibits inhibitory activity for the enzyme adenine deaminase. It also shows antitumor and immunosuppressive activity. Pentostatin has been registered as a drug from the group of chemotherapeutic agents, indicated in the treatment of cancer in oncology and hematology hospital departments [57,58]. Cordymin is a compound found in fruiting bodies of *C. militaris* and is classified as a peptide with antimicrobial activity [59].

Fruiting bodies of *C. militaris* contain water-soluble vitamins (vitamin B2, vitamin B3, and vitamin C) and fat-soluble vitamins (vitamin A and vitamin E) [9]. Vitamin B2 (riboflavin) and vitamin B3 (niacin) reduce the feeling of tiredness and fatigue and are responsible for normal energy-yielding metabolism. Vitamin C protects cells from oxidative stress and contributes to maintain the normal function of the immune system, support collagen formation and increase the absorption of iron from the gastrointestinal tract. Vitamin A plays a role in the process of cell specialization and maintenance of normal vision. Vitamins E protects cells against oxidative stress [60].

Bioelements such as magnesium, potassium, selenium and sulfur have been detected in fruiting bodies of *C. militaris*. The high concentration of these bioelements in *C. militaris* makes it an alternative source of these minerals for human diet. The content of minerals present in *C. militaris* is summarized in Table 1. The concentration of biominerals such as boron, manganese, copper and iron is relatively negligible and ranges from 5 to 31 mg/kg dry weight [7].

In fruiting bodies of *C. militaris*, selenium occurs in an organic form and is bound to amino acids or proteins. Selenium is chelated with the amino acid L-methionine (selenomethionine) or L-cysteine (selenocysteine). It combines with proteins to form methylselenocysteine. The addition of selenium (sodium selenate) to *C. militaris* medium increases the concentration of active compounds in fruiting bodies—nucleosides, polysaccharides, amino acids, and organic selenium [61,62]. Enrichment of the *C. militaris* substrate with organic or inorganic selenium (sodium selenate(IV); sodium selenate(VI)) increases the concentration of cordycepin and adenosine in fruiting bodies [63]. Selenium-fortified polysaccharide was obtained from the mycelium of *C. militaris* (SeCSP -I). An in vitro experiment demonstrated the antioxidant activity of SeCSP-I [19].

## 4. Biological Activity

Scientific studies have shown significantly more diverse biological activities of *C. militaris* than those of *C. sinensis*. *C. militaris* has been shown to exert the following activities: ergogenic, immunostimulating, antitumor, antioxidant, anti-inflammatory, antiviral, neuroprotective and hypolipemic [11,14].

### 4.1. Ergogenic and Anti-Fatigue Activity

The activity of reducing the feeling of fatigue correlates with the strengthening and ergogenic effect, i.e., the improvement of physical performance. Cordycepin is the main bioactive compound with ergogenic potential in *C. militaris*. The ergogenic activity of cordycepin is related to its physiological function as indirect precursor of ATP and NO [11,12].

Cordycepin, similar to creatine, is indirect precursor of ATP. The ergogenic activity of creatine is supported by numerous pieces of scientific evidence. Creatine increases physical performance in successive bursts of short-term, high intensity exercise [64]. In a group of 11 men, who were physically active recreationally, immediate consumption of ATP at a dose of 400 mg/day contributed to the improvement of physical performance and oxygen and energy consumption in lower body exercises [65].

#### In Vivo Research 

In experiments conducted on rodents, the anti-fatigue activity of *C. militaris* was demonstrated. In the swimming test, there was a prolongation of time to fatigue in mice. The reduction in fatigue caused by *C. militaris* in experimental animals was due to an increase in the concentration of ATP and antioxidant enzymes, a decrease in the concentration of lactate and ROS, and activation of the 5ʹ-AMP-activated kinase (AMPK) and AKT/mTOR pathways [66,67,68]. Mice fed with the extract of *C. militaris* with cordycepin showed an improvement in exercise performance in a grip strength test [69].

Hirsch et al. [70] demonstrated an improvement in exercise performance in recreationally active subjects following the administration of 4 g/day of a mushroom mixture (containing *C. militaris*) for 3 weeks. The effect of mushroom mixture with *C. militaris* on exercise capacity is associated with an improvement in maximum oxygen consumption (VO_2max_), an increase in time to exhaustion, and a reduction in lactic acid concentration.

Hirsch et al. [71] verified the effect of using a mushroom mixture (containing *C. militaris*) on physical performance during high-intensity exercise after 1 week and 3 weeks of supplementation. Twenty-eight recreationally active people in aged 18–27 years were subjected to this experiment. VO_2max_, time to exhaustion and ventilation threshold (VT) were measured during an exercise test on a cycling ergometer. The intake of the mushroom mixture with *C. militaris* improved the tolerance of effort in the performed exercises. In another study, Dudgeon et al. [72] demonstrated that the consumption of a mushroom mixture with *C. militaris* in a group of healthy subjects aged 19–34 years contributed to an improvement in physical endurance, an increase in time to exhaustion, an increase in VO_2max_ and a decrease in blood lactate concentration. In the aforementioned scientific works, *C. militaris* was a component of a mixture of adaptogenic mushrooms (trade name PeakO_2_^®^), which also contained *Ganoderma lucidum*, *Pleutorus eryngii*, *Lentinula edodes*, *Hericium erinaceus* and *Trametes versicolor* [73].

### 4.2. Immunostimulating Activity

Extracts from fresh *C. militaris* fruiting bodies exhibit greater immunostimulatory activity (in vitro and in animal model) than those from dried fruiting bodies. There was no significant difference in the content of cordycepin and adenosine between fresh and dried raw material, but a significant discrepancy was noted in the concentration of polysaccharides, polyphenols and flavonoids, which were greater in fresh *C. militaris* [74].

It has been proven that, depending on the solvent used and thus the active substances contained in the *C. militaris* extract, there are two pathways to stimulate the immune system function. Water or ethanol extract (50%) of *C. militaris* containing polysaccharides stimulates type 1 immune response, whereas ethanol extract (70–80%) containing cordycepin stimulates type 2 immune response [75].

#### 4.2.1. In Vitro Research

*C. militaris* extract containing cordycepin exhibits immunostimulatory effects on mouse macrophages. The mechanism of the immunostimulatory activity of *C. militaris* and cordycepin is based on the activation of macrophages to produce NO and proinflammatory cytokines IL-1β, IL-6, TNF-α and prostaglandin-2 (PGE2), as well as an increase in the activity of induced nitric oxide synthase (iNOS) and cyclooxygenase-2 (COX-2). The stimulation of macrophages to produce proinflammatory mediators was due to the activation of the nuclear transcription factor (NF-κB) by *C. militaris*/cordycepin [11,12].

Another experiment found that the CMP-W1 polysaccharide could stimulate the proliferation of lymphocytes [20]. The novel low-molecular-weight polysaccharide (CMPB90-1) isolated from *C. militaris* was shown to have immunostimulatory activity. In in vitro tests, the production of lymphocytes and NK cells and stimulation of phagocytosis by macrophages were demonstrated in mouse spleen cells treated with *C. militaris* polysaccharide [22].

It was shown that the CPMN Fr III polysaccharide (β-1,4-branched-β-1,6-galactoglucomannans) obtained from the cultured mycelia of *C. militaris* induced the production of NO, IL-1β and TNF-α by macrophage cells [76]. Furthermore, it was demonstrated that the polysaccharides found in fruiting bodies of *C. militaris* induced an immune response in murine macrophages. The experiment showed the intensification of the activity of NO, and TNF-α [77]. A novel polysaccharide (signed as PLCM) isolated from culture broth of *C. militaris* can induce immune response via MAPK and NF-κB signaling pathways [78].

#### 4.2.2. In Vivo Research

Tests on rodents showed an increase in phagocytosis and an increase in the production of lymphocytes in spleen cells. The immunostimulatory activity was explained by the action of polysaccharides present in the tested mushroom material [14]. Immunostimulatory activity, through macrophage activation, was confirmed for the polysaccharide fraction from fruiting bodies of *C. militaris* [77].

Jeong et al. [79] demonstrated the immunomodulatory activity of cordycepin and *C. militaris*, which was associated with the inhibition of tumor growth in experimental mice. The study showed an increase in the production of IL-4, a decrease in the concentration of IL-2 and TGF-β and an increase in the level of T lymphocytes (CD4 and CD8).

In studies on healthy adult males supplemented with 1.5 g/day of *C. militaris* (in capsules) for 4 weeks, the immunostimulatory activity was found to be enhanced due to an increase in the levels of IL-2, IL-12, NK, TNF-α and IFN-γ [80].

Another experiment analyzed the effect of a 12-week supplementation of *C. militaris* on the course of upper respiratory tract infection and the immune response in a group of 100 patients aged 20–70 years. The study showed that the use of *C. militaris* did not have a significant effect on the frequency and symptoms of colds. However, an increase in NK cell activity and an increase in IgA concentration were observed, which indicated the immunostimulatory effect of *C. militaris* [81].

### 4.3. Antitumor Activity

#### 4.3.1. In Vitro Research

In a study by Yoo et al. 2004 [82], the antitumor activity of *C. militaris* extract was confirmed. The cytotoxic activity of *C. militaris* aqueous extract against human breast cancer cells have been demonstrated. The promotion of apoptosis in tumor cells resulted from the activation of caspase-3 [83].

In scientific work by Park et al. [84], the authors demonstrated that *C. militaris* aqueous extract inhibits proliferation and induces apoptosis in human lung cancer cells. The antitumor activity of *C. militaris* was associated with an increase in the enzymatic activity of caspase-3, caspase-8 and caspase-9, and inhibition of the telomerase enzyme. The study reported an increase in the concentration of the Fas protein, which is associated with the “death receptor” of cancer cells.

Cordycepin and ergosterol isolated from *C. militaris* have shown antiproliferative activity against human colon cancer cells. Proliferative activity was associated with anti-inflammatory activity [85].

The in vitro cytotoxic activity of cordycepin extracted from *C. militaris* was tested. The results revealed that cordycepin inhibits the proliferation and migration of human bladder cancer cells. The cytotoxic activity of cordycepin has been linked with its effect on various signaling pathways, namely metalloproteinase-9 (MMP-9), TNF-α, NF-κB and protein activator-1 (AP-1) [86].

The apoptotic activity of *C. militaris* against glioblastoma tumor cells was also confirmed, and the *C. militaris* extract was shown to restrict the angiogenesis of human malignant melanoma cells [87,88]. In another study, the CMPS-II polysaccharide in the extract was shown to inhibit the growth of lung cancer cells [89].

A more recent study showed that cordycepin suppressed the growth of human liver cancer cells. The mechanism of antitumor activity of cordycepin has been linked with a reduction in the expression of the chemokine CxCR4, which promotes invasiveness and migration of liver cancer cells [90].

*C. militaris* induces apoptosis in ovarian cancer cells. The mechanism of antitumor activity was shown to involve the activation of TNF-α, TNFR1, NF-κB, caspase-3 and caspase-9, and reduction in the levels of Bcl-2 and BclxL [91].

*C. militaris* has been shown to induce apoptosis of cancer cells in non-small cell lung cancer. The basis of apoptotic activity its associated with inhibition of tectonic-3 protein (TCTN3) expression. Reduction in TCTN3 expression correlated with the suppression of different signaling pathways such as: Smoothened (SMO), Patched1 (PTCH1) and glioma-associated (GLI1) pathways. The aforementioned processes correlate with decreased Bcl-2, BclxL, and increased Bak, cleaved caspase-3 and caspase-9 levels [92].

#### 4.3.2. In Vivo Research

Park et al. [84] reported that the aqueous extract of *C. militaris* reduced the mass and volume of the tumor and also extended the survival time (viability) of mice with lung cancer. Similar results were obtained by Jeong et al. [79], wherein *C. militaris* additionally enriched with cordycepin inhibited tumor growth and increased the viability of tumor-bearing mice. The study showed the immunotropic effect of cordycepin, which correlated with its antitumor activity.

### 4.4. Antioxidant Activity

The antioxidant activity was confirmed mainly for polysaccharides present in *C. militaris*. Few studies have shown the antioxidant properties of cordycepin [93]. The antioxidant activity of *C. militaris* may also be influenced by other chemical constituents present in fruiting bodies, e.g., ergothioneine, phenolic compounds, carotenoids and selenium [7,8,9].

#### 4.4.1. In Vitro Research

The antioxidant potential of *C. militaris* in terms of inhibiting lipid peroxidation exceeds that of *C. sinensis*. The antioxidant activity was related to the content of polysaccharides and phenolic compounds in *C. militaris* fruiting bodies [94]. Previous studies confirmed the ability of the polysaccharide P70-1 and CBP-1 obtained from *C. militaris* to eliminate hydroxyl radicals [95,96]. The antioxidant activity and the ability to chelate iron ions (Fe2+) were also proven for the polysaccharide designated as CM-hs-CPS2 [97].

In many in vitro experiments, the antioxidant activity of the polysaccharide fractions was observed for the components WCBP50, CMP, CMP-1 and SeCSP-I [16,19,21,98]. The addition of selenium to the medium of *C. militaris* was found to increase the antioxidant activity of the polysaccharide fractions [63].

#### 4.4.2. In Vivo Research

Experiments on rodents fed *C. militaris* containing polysaccharides showed an increase in the activity of enzymes with antioxidant potential, namely superoxide dismutase (SOD), catalase and GPX, and/or a reduction in the level of malondialdehyde (MDA) [14,99]. Antioxidant activity was also observed in other experiments on rodents and correlated with the neuroprotective or hepatoprotective activities, described in subsequent sections of this review.

### 4.5. Anti-Inflammatory Activity

#### 4.5.1. In Vitro Research

In a model of lipopolysaccharide (LPS)-induced inflammation in macrophages, cordycepin was shown to reduce the expression of TNF-α, COX-2, iNOS and NF-κB [100]. In subsequent in vitro tests, the anti-inflammatory activity of *C. militaris* was confirmed to result from the inhibition of production of proinflammatory mediators, namely NO, TNF-α and IL-6, which were induced by LPS in murine macrophages. The group of bioactive compounds of *C. militaris* was not defined in the study [101].

It was proven through in vitro experiment that cordycepin and ergosterol from *C. militaris* inhibit the release of inflammatory mediators: NO, TNF-α and IL-12, which correlates with antiproliferative activity in colon tumor cells [85].

In vitro experiments using mouse microglia demonstrated that cordycepin (from *C. militaris*) inhibits the activity of COX-2 and iNOS enzymes and lowers the levels of inflammatory mediators: NO, TNF-α, PGE2 and IL-1β. Cordycepin was also shown to inhibit the activity of NF-κB and to inhibit the phosphorylation of mitogen-activated protein kinases (MAPKs). These results confirmed the neuroprotective effect of cordycepin and *C. militaris* extracts and have opened up new opportunities for further use of *C. militaris* in research and therapy of neurodegenerative diseases [102].

#### 4.5.2. In Vivo Research

In an experimental mice model of colitis induction, the administration of *C. militaris* was shown to contribute to the inhibition of the activity of iNOS and TNF-α and the reduction in mast cell degranulation. The study, however, did not identify a specific group of bioactive compounds responsible for anti-inflammatory activity [103].

In scientific work Won and Park [104] was confirmed in vivo to ability of cordycepin contained in mycelium *C. militaris* to inhibit the activity of iNOS and reduce the concentration of NO in inflammatory reaction. The study also demonstrated analgesic activity.

In a study conducted on mice, Smiderle et al. [105], concluded that a polysaccharide with a linear structure of β-(1 → 3)-D-glucan obtained from *C. militaris* exhibited anti-inflammatory and antinociceptive activity. The mechanism of the biological activity of this polysaccharide resulted from its inhibition of COX-2, TNF-α and IL-1β activity. The anti-inflammatory and antinociceptive activity of the polysaccharide was similar to that of the reference compounds from the group of non-steroidal anti-inflammatory drugs (acetylsalicylic acid, indomethacin and diclofenac).

Cerebrosides, a new group of compounds, were indicated as a component with anti-inflammatory activity. The mechanism of suppression of the inflammatory process was associated with the inhibition of expression of COX-2, iNOS, NF-κB, IL-1β and IL-6 [106].

Cai et al. and Gai et al. [107,108] confirmed the possibility of using *C. militaris* for treating chronic bronchitis in humans.

### 4.6. Hypoglycemic Activity

In experiments on rats, *C. militaris* extract was found to influence the use of glucose by tissues and reduce insulin resistance. In another study on rats, the hypoglycemic activity of *C. militaris* extract containing polysaccharides was demonstrated [14]. Furthermore, the ability of the LCMPs-II polysaccharide to inhibit the α-glucosidase enzyme was demonstrated by Zhu et al. [18]. In contrast, the cerebroside fraction inhibited PTP1B activity [109]. Exopolysaccharide (EPS III) isolated from culture broth of *C. militaris* demonstrated hypoglycemic activity in streptozotocin-induced diabetic mice. EPS III showed antagonist activity against the enzyme on α-glucosidase with the dose–effect relationship [110].

In an animal model on diabetes, 21 days of treatment with cordycepin (from *C. militaris*) in alloxan-induced diabetic mice contributed to the improvement of many symptoms of metabolic syndrome, including regulation of glucose tolerance and glucose absorption, and decreased blood glucose concentration [111]. Current scientific studies also indicate the potential of cordycepin in the treatment of diabetes by the regulation of expression liver proteins such as Nfat3, Flcn and Psma3. These proteins are correlated with generation of energy (ATP), AMPK signaling pathway and ubiquitin proteasome system (UPS) [112].

### 4.7. Antimicrobial Activity

Cordymin is a peptide isolated from *C. militaris*. It was shown to have antifungal and antiviral activity in in vitro tests. Cordymin inhibited the growth of various fungal species, including *Bipolaris maydis* and *Candida albicans*. It also inhibited reverse transcriptase of human immunodeficiency virus (HIV). Several scientific studies have confirmed the anti-inflammatory and antinociceptive activity of cordymin [59].

In vitro studies have demonstrated the antiviral activity of cordycepin and its derivatives. The antiviral activity was confirmed for the influenza virus, Epstein–Barr virus (EBV), herpes simplex virus (HSV) and HIV. The mechanism of antiviral activity of cordycepin is related to the inhibition of reverse transcriptase and RNA polymerase of the virus [113,114,115].

In the United States in 2020, the Food and Drug Administration (FDA) repurposed potential and investigational drug or molecule candidates and included cordycepin as a chemical structure with antiviral activity. Precursor research in India on cordycepin created new possibilities for the use of this molecule in the treatment of COVID-19. Cordycepin exhibited strong chemical interactions with SARS-CoV-2, with receptor-binding domain (RBD) at the spike protein (S) and the main protease (M^pro^) of the virus in in silico study. Anti-SARS-CoV-2 mechanisms of the action of cordycepin correlated with inhibited viral replication [116]. Another in silico study using pharmacological and molecular modeling, simulation and estimation indicated that cordycepin is the potential inhibitor for RNA-dependent RNA polymerase (RdRp) of SARS-CoV-2 [117].

### 4.8. Effect of C. militaris on the Endocrine System

*C. militaris* has been shown to stimulate hormonal activity in rodents. There has been an increase in testosterone levels in plasma [118,119].

Cordycepin has been shown to stimulate steroidogenesis in rodents and increase testosterone and progesterone levels [120].

Sohn et al. [121] showed that the administration of cordycepin (from *C. militaris*) in a group of experimental rats improved testicular function, stimulated spermatogenesis and increased sperm motility. The study also showed that cordycepin administration increased calcium concentration and decreased urea and creatinine levels in the blood of rats. Similar results in terms of improvement in sperm quality were reported in rodent studies by Kopalli et al. [122].

Improvement of quality parameters of semen in infertile boars to which *C. militiaris* mycelium was given has been demonstrated. The improvement was connected with the increase in the amount of sperm and its motility [123].

In vitro and in vivo experiments on rodents revealed that the *C. militaris* extract containing cordycepin reduced the proliferation of prostate cells and regulated the concentration of androgens [124].

Research concerning the effect of *Cordyceps* spp. or cordycepin on the endocrine system of human objects are limited. The promising effects of *Cordyceps* spp. or cordycepin on steroidogenesis in rodents have not yet been confirmed in humans. In a group of young male adults, it has been proven that 8-week supplementation of 2.4 g per day of *C. sinensis* (containing 5.92 µmol/g of adenosine, 1.23 µmol/g of cordycepin and 8.81 µmol/g of ergosterol) does not significantly affect testosterone levels in volunteers. So far, no studies have been conducted to analyze the effect of *C. militaris* on testosterone concentration in men. One can only speculate in the area of potential results due to the fact that *C. miliatris* presents a higher concentration of cordycepin than *C. sinensis* [125].

### 4.9. Effect of C. militaris on the Respiratory System

In vitro experiments provided evidence that cordycepin from *C. militaris* affects the transport of sodium, potassium and chloride ions in the epithelial cells of the respiratory tract [126].

Early publications found that extracts from fruiting bodies of *C. militaris* are less effective than the reference compounds—prednisolone and montelukast—in reducing the inflammatory process in a mouse model of asthma. *C. militaris* extracts showed negligible effects on the reduction in IgE concentration, eosinophilia and inhibition of leukotriene synthesis [127]. However, later scientific works demonstrated in an asthma model in mice that cordycepin alleviates airway hyperresponsiveness, reduces inflammation and decreases IgE and eosinophil levels. A decrease in the expression of IL-4, IL-5, IL-13 and NF-κB was observed [12].

Cordycepin was also shown to reduce airway remodeling in an asthma model in rats. A decrease in the levels of IgE, eosinophils and neutrophils and a decrease in the expression of TNF-α, TGF-β1, IL-5 and IL-13 were also observed [128].

The polysaccharide, designated as CPS, reduced IgE levels in a mouse model of asthma, inhibited cell proliferation and infiltration and alleviated inflammation and airway hyperresponsiveness. CPS also inhibited the secretion of IL-4, IL-5, IL-13 and IFN-γ and reduced the expression of TGF-β1 [129].

In human studies by Cai et al. and Gai et al. [107,108] some efficacy of *C. militaris* was demonstrated in the treatment of chronic bronchitis. Jung et al. [81] found that although *C. militaris* did not have a significant effect on the course of upper respiratory tract infection in a group of volunteers, an immunostimulatory effect was observed.

### 4.10. Effect of C. militaris on the Locomotor System

Scientific research indicated the potential benefits of *C. militaris* in the treatment of osteoporosis. An in vitro experiment showed that *C. militaris* inhibited osteoclast differentiation and reduced the expression of genes encoding this process [130]. It has been proved that cordycepin exhibits anti-inflammatory activity in human osteoarthritis chondrocytes. Cordycepin inhibited the production of PGE2 and NO, and decreased the expression of NF-κB, induced by IL-1β [131].

In an inflammatory-induced osteoporosis model in experimental rats, cordycepin showed anti-inflammatory activity and limited bone loss [132].

Cordycepin and *C. militaris* inhibited osteoclast differentiation in in vitro tests. The mechanism of their biological activity has been linked to the inhibition of the NF-κB ligand receptor activator (RANKL). In addition, the mRNA expression of genes related to osteoclastogenesis (TRAP, Cathepsin K, MMP-9 and NFATc1) was also inhibited by CME and cordycepin. In addition, cordycepin significantly inhibited RANKL-induced p38 and NF-κB phosphorylation. Moreover, in a mouse model of LPS-induced osteoporosis, cordycepin and *C. militaris* limited bone loss [133]. In an experimental model of osteoarthritis in rats, cordycepin as a polyadenylation inhibitor was shown to reduce pain and inflammation in the synovium [134].

### 4.11. Effect of C. militaris on the Nervous System

The neuroprotective effect of cordycepin was confirmed in an in vitro study on mouse microglia cells. The protective effect of cordycepin on neurons was dependent on its anti-inflammatory activity [102] or antioxidant activity [135].

Lee et al. [136] proved that *C. militaris* antagonized the scopolamine-induced memory deficit effect in experimental rats. The neuroprotective effect of *C. militaris* was also confirmed in a rodent model of dementia and ischemic brain damage [137]. Yuan et al. [138] showed that a polypeptide isolated from *C. militaris* improves memory in mice. Decreased activity of acetylcholinesterase (AChE) and enhanced GABA neurotransmission were also detected. Field work also showed antioxidant and neuroprotective activity, increased SOD activity and reduced MDA concentration. In a model of Alzheimer’s disease induced by the β-amyloid protein (Aβ1-42) in experimental mice, *C. militaris* improved procognitive functions in the following tests: water maze, new route and object recognition tests [135]. In rodents treated with doxorubicin chemotherapy, *C. militaris* extract was shown to reduce chemotherapy-induced oxidative stress. Postmortem of the experimental rodents showed that the *C. militaris* extract increased the concentration of ATP in the brain and inhibited the activity of AChE [139].

One study verified the effect of supplementation of a mushroom beverage (containing *C. militaris* with cordycepin, polysaccharides and mannitol) on the regulation of emotions in humans. The potential effect on alleviating the symptoms of depression has not been confirmed so far; that is, the results of the research have not been published yet [140].

### 4.12. Effect of C. militaris on the Cardiovascular System

Cordycepin is a candidate for development of potential antiatherosclerotic agents, because it improves vascular responses in smooth muscle cells [141].

In experiments with rodents, cordycepin from *C. militaris* has been shown to lower triglycerides, total cholesterol, low-density lipoprotein (LDL) and very low-density lipoprotein (VLDL) level, in an animal model of hyperlipidemia. Cordycepin showed the features of an AMPK activator and an inhibitor of lipoprotein and hepatic lipase [142,143].

Cordycepin inhibited adipogenesis and lipid deposition in adipocytes in in vitro experiments. The mechanism of the biological activity of cordycepin was found to be associated with the suppression of the C/EBPβ, PPARγ and mTORC1 pathways and activation of AMPK [144]. The hypolipidemic activity of cordycepin and its ability to activate AMPK, the γ1 region, were also confirmed in in vitro tests [145].

An in vitro experiment and an in vivo study in rodents also confirmed the hypolipidemic activity of the polysaccharide fractions isolated from *C. militaris* [146,147]. Similarly, Huang et al. [148] demonstrated the hypolipidemic activity of intracellular (IPCM) and extracellular polysaccharides (EPCM) from *C. militaris* in mice fed a high-fat diet.

An in vitro experiment confirmed the antiaggregation activity of cordycepin isolated from *C. militaris*. Inhibition of aggregation of human platelets resulted from an increase in the activity of cyclic guanosine-3ʹ,5ʹ-monophosphate (cGMP) and cyclic adenosine-3ʹ,5ʹ-monophosphate (cAMP), a decrease in the intracellular concentration of calcium ions and inhibition of the synthesis of thromboxane A2 (TXA2) by cordycepin [149]. Cordycepin exhibited cardioprotective activity in an isolated ischemic rat heart [150].

In an in vivo experiment conducted in mice, the sodium nitrite-induced toxicity test and the ischemia test showed that the fraction of CMN1 polysaccharides isolated from fruiting bodies of *C. militaris* prevented hypoxia [17].

*C. militaris* extract enriched with cordycepin inhibited collagen-induced platelet aggregation in in vitro tests. The antiaggregation activity resulted from the inhibition of fibrinogen attachment to glycoprotein IIb/IIIa, as well as the stimulation of VASP phosphorylation, inhibition of PI3K/A-kinase phosphorylation and increase in cAMP concentration [151].

Further in vitro and in vivo experiments demonstrated the neuroprotective effect of *C. militaris* (WIB801C) extract with a quantified cordycepin concentration. Cordycepin reduced the ischemic area, reduced brain swelling and restricted damage to the blood–brain barrier in experimental rats. The neuroprotective effect was related to the anti-inflammatory activity of cordycepin [152]. More recent studies indicate greater antiplatelet activity than anticoagulant activity of *C. militaris* [153].

### 4.13. Other Biological Activity of C. militaris

In in vitro experiments, *C. sinensis* showed a greater ability than *C. militaris* to affect the gut microbiota [154].

In rodent experiments, *C. militaris* polysaccharides minimized damage to hepatocytes. Hepatoprotective activity of *C. militaris* extract correlated with its antioxidant and hypolipemic activity [147]. In a study conducted in patients with impaired liver function, supplementation with 1.5 g/day *C. militaris* (cordycepin) for 8 weeks protected the liver against lipid accumulation [155].

The nephroprotective activity of *C. militaris* was demonstrated by inhibiting the hypertrophy of glomerular cells induced by LDL in in vitro [156]. The limitation of proliferation on glomerular mesangium cells was confirmed for *C. militaris* extracts [157]. The nephroprotective effect of *C. militaris* or cordycepin was confirmed in a diabetic animal model in mice [111,158].

## 5. Safety Assessment and Toxicology

In vivo experiments showed that high-dose administration of *C. militaris* (4.9 × 10^8^ spores/mouse) for 7 days in rodents did not induce toxic effects. No significant differences were observed between the control group and the experimental group fed with *C. militaris* in terms of body weight, blood hematological parameters, and food and water consumption [159].

In experimental rats, no toxic effect of *C. militaris* was observed at the dose of 3000 mg/kg/day in the sub-acute oral toxicity test. No genotoxic and teratogenic effects were found. In the sub-chronic toxicity test (90-day) in experimental rats fed with *C. militaris*, no changes were found in clinical, histopathological and hematological parameters. The no-observed-adverse-effect level (NOAEL) was determined for *C. militaris* mycelium at 4000 mg/kg/day in experimental rats [160]. The maximum tolerance dose of cordycepin has been confirmed at 3600 mg/kg/day in mice without any adverse effects [111].

In human studies, *C. militaris* supplementation did not correlate with the occurrence of serious adverse events. Few of the reported mild adverse events were associated with gastrointestinal disturbances (Table 2).

Beauveriolides and militarinones are among the compounds present in *C. militaris* with cytotoxic potential. Pentostatin and cordycepin have been shown to exert gastrotoxic effects. Pentostatin is also toxic to the lungs and nervous system [57].

Bai and Sheu [161] isolated an 18-kDa nonglycosylated protein (CMP) that induces mitochondrial-dependent apoptosis in murine cells. CMP is unstable and degrades in a high temperature or alkaline environment. Heat treatment of *C. militaris* is therefore necessary to deactivate CMP and further process the *C. militaris* raw material for food, dietary supplements and medicines.

Heavy metals such as cadmium, lead, mercury and arsenic pose toxicological hazards to human health. Although these metals may be accumulated in the fruiting bodies of various species of fungi, these do not occur in mushrooms used for cultivation. A quantitative analysis of heavy metals in fruiting bodies of wild-growing *C. militaris* showed that lead concentration did not exceed 0.2 mg/kg dry weight, whereas mercury concentration did not exceed 0.5 mg/kg dry weight. Relatively high levels of arsenic and cadmium were noted, amounting to 0.4 mg/kg and 0.2 mg/kg dry weight, respectively [7,162].

## 6. Conclusions

Strengthening endurance of the human body includes stimulation of immune functions and reduction in fatigue. To summarize, the presence of several bioactive compounds in edible fungi such as *C. militaris* makes them worthy to be considered as a functional food. The inclusion of *C. militaris* extract in the diet can have several health benefits; moreover, the consumption of many mushroom species that can be grown on household waste will popularize the mushroom sector in the sustainable economy of many regions of the world. At the same time, a large number of bioactive compounds hampers clinical trials and creates problems for designing drugs based on mushroom extracts. A single substance, however, may have a less therapeutic effect and may not show synergistic effects with other compounds present in mushroom fruiting bodies. Among all bioactive substances mentioned in the studies, the greatest therapeutic potential is associated with cordycepin. Recent scientific reports indicate the potential of cordycepin as antiviral agent against SARS-CoV-2 in COVID-19.

## Figures and Tables

**Figure 1 foods-10-02634-f001:**
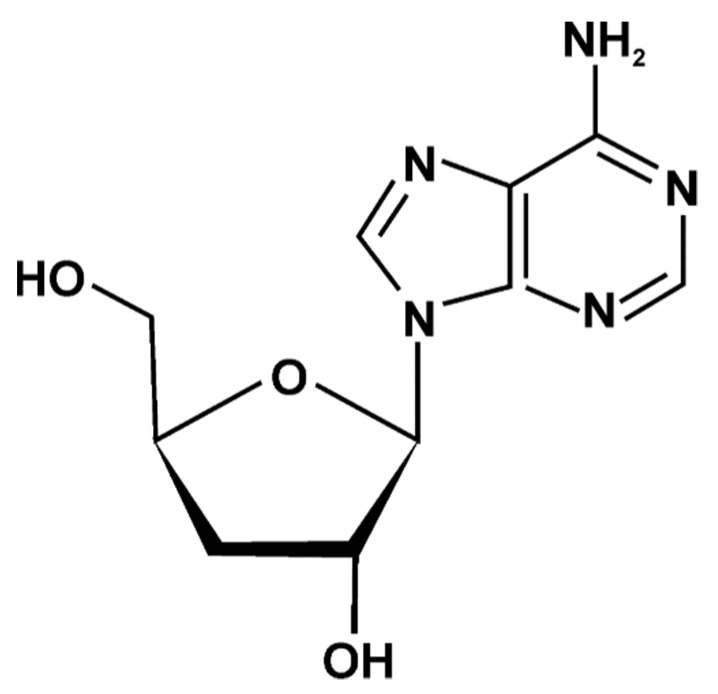
Chemical structure of cordycepin.

**Figure 2 foods-10-02634-f002:**
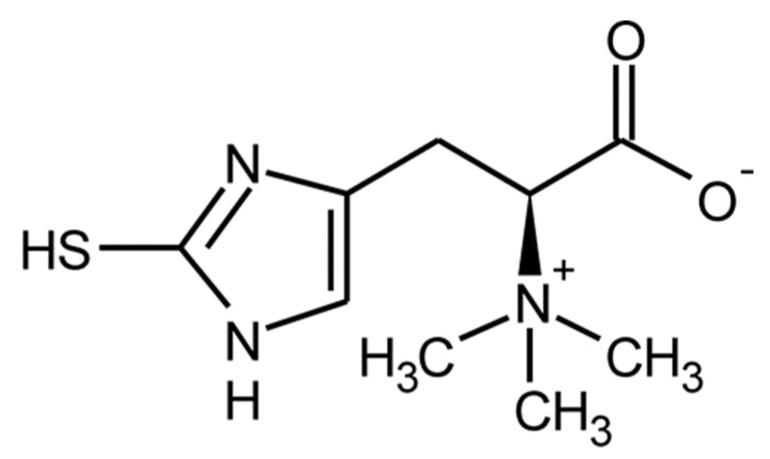
Chemical structure of ergothioneine.

**Figure 3 foods-10-02634-f003:**
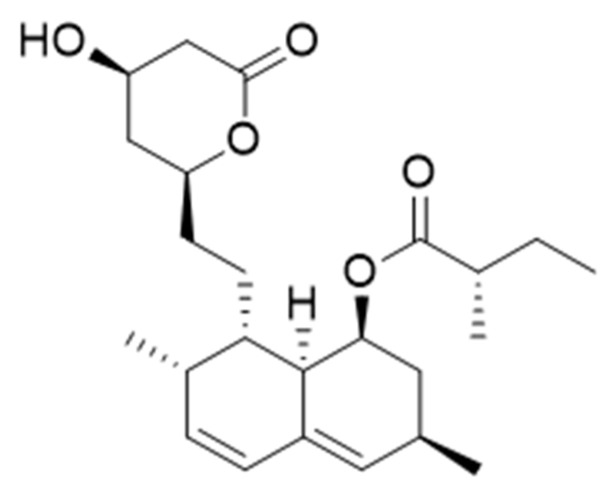
Chemical structure of lovastatin.

**Table 1 foods-10-02634-t001:** Content of bioactive compounds and nutrients present in the mycelium and fruiting bodies of *C. militaris*.

	Content		
Bioactive Compound	Mycelium	Fruiting Bodies	References
Cordycepin	1.82 mg/g	1.10 mg/g	[9]
Cordycepin	1.74 mg/g	5.28 mg/g (Water extrct)	[7]
Cordycepin		8.37 mg/g (Ethanol extract)	[7]
D-mannitol	5.2 mg/kg	4.7 mg/kg	[9]
Ergothioneine	130.6 mg/kg	782.3 mg/kg	[9]
Ergothioneine	123.4–785.1 mg/kg	409.8 μg/g	[7,8]
GABA	68.6–180.1 mg/kg	756.30 μg/g	[7,8]
Lovastatin	37.7–57.3 mg/kg	2.76 μg/g	[7,8]
**Vitamins**			
Vit. A	100 mg/kg	96 mg/kg	[9]
Vit. E (tocopherols)	1.3 mg/kg	3.6 mg/kg	[9]
Vit. B2 (riboflavin)	0.32 mg/kg	0.16 mg/kg	[9]
Vit. B3 (niacin)	15.2 mg/kg	4.9 mg/kg	[9]
Vit. C	Not detected	<2 mg/kg	[9]
**Bioelements**			
Magnesium	3414 mg/kg	4227 mg/kg	[9]
Sulfur	2558 mg/kg	5088 mg/kg	[9]
Potassium	12,183 mg/kg	15,938 mg/kg	[9]
Selenium	<0.5 mg/kg	0.4 mg/kg	[9]
Iron	9 mg/kg	31 mg/kg	[9]
Calcium	11 mg/kg	797 mg/kg	[9]
Zinc	10 mg/kg	Not detected	[9]
**Nutrients**			
Protein	39.5%	59.8%	[9]
Protein	Not analyzed	29.7%	[7]
Fat	2.2%	8.8%	[9]
Fat	Not analyzed	2.9%	[7]
Carbohydrate	39.6%	29.1%	[9]
Carbohydrate	Not analyzed	54.3%	[7]

**Table 2 foods-10-02634-t002:** Details of *C.*
*militaris* research on human participants.

Research Group	Quantity of Raw Material *C.* *militaris*	Supplementation Time	Biological Activity	Bioactive Compound	Adverse Events	References
10 healthy individuals of both sexes aged 19–24 years	4 g/day ^1^	3 weeks	Ergogenic	Undefined	Undefined	[70]
28 healthy individuals of both sexes aged 18–27 years	4 g/day ^1^	1–3 weeks	Ergogenic	Undefined	Undefined	[71]
43 healthy individuals of both sexes aged 19–34 years	1–12 g/day ^1^	1–4 weeks	Ergogenic	Undefined	Gastrointestinal	[72]
79 healthy males aged 19–64 years	1.5 g/day	4 weeks	Immunostimulation	Cordycepin	No serious adverse events	[80]
100 healthy volunteers aged 20–70 years	1.5 g/day	12 weeks	Immunostimulation	Cordycepin	No serious adverse events	[81]
510 patients with chronic bronchitis	Undefined ^2^	Undefined	Anti-inflammatory	Undefined	Undefined	[107]
425 patients with chronic bronchitis	3 g/day ^2^	8 weeks	Anti-inflammatory	Undefined	Undefined	[108]
57 patients with mild hepatic impairment aged 31–61 years	1.5 g/day	8 weeks	Hepatoprotective	Cordycepin	No serious adverse events	[155]

^1^ Adaptogenic mushroom mixture (trade name: PeakO_2_^®^) ^2^ Access to publications only as abstract.

## Data Availability

Not applicable.

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
