# Peer review of "Cordyceps militaris: An Overview of Its Chemical Constituents in Relation to Biological Activity"

_foods, 2021, doi:10.3390/foods10112634_

Round 1
Reviewer 1 Report
I reviewed the manuscript entitled, Cordyceps Militaris: An Overview of Its Chemical Constituents in Relation to Biological Activity. The manuscript is well written. However, authors should focus in detail for some aspects of the review.
Abstract should be revised with review objectives and findings of the review
Table 1. Please insert column and describe their fruit parts like extracted parts.
Sections 2.4, 2.6, and 2.8 should be described in details with quantity and comparison with other species
Sections 3.6 to 3.8 should be described in details
Citation format, for example Sohn et al. 2012 [99] and many other are not the correct format. Please revise throughout the manuscript
Short sentences should be combined throughout the manuscript
No proper discussion for many sections, for example sections like 3.9, 3.10, 3.11 etc
References should be revised according to journal format. Many scientific names should be in Italics.
Other suggestions are attached (PDF)

Author Response
Dear Editor,
Thank you very much for reviewing our manuscript. We also thank the Reviewers for carefully reading our manuscript and providing useful comments, which were a great help to us. Based on the Reviewers’ comments, we have revised the manuscript. Bellow our response for Reviewer 1 suggestions:
I reviewed the manuscript entitled, Cordyceps Militaris: An Overview of Its Chemical Constituents in Relation to Biological Activity. The manuscript is well written. However, authors should focus in detail for some aspects of the review.
Dear Reviewer thank you for carefully reading our manuscript and providing useful comments, which were a great help to us to improve the MS. Based on yours comments, we have revised the manuscript.:
Abstract should be revised with review objectives and findings of the review – We revised abstract section.
Table 1. Please insert column and describe their fruit parts like extracted parts. – We condensed Table 1 and Table 2 and described their fruit parts like extracted parts.
Sections 2.4, 2.6, and 2.8 should be described in details with quantity and comparison with other species – The aforementioned sections have been described in details and developed. Section “Vitamins” and “Bioelements” have been condensed and described in one chapter “Other bioactive compounds in C. militaris fruiting bodies”.
Sections 3.6 to 3.8 should be described in details – The aforementioned sections have been described in details and developed.
Citation format, for example Sohn et al. 2012 [99] and many other are not the correct format. Please revise throughout the manuscript – We changed to the correct citation format.
Short sentences should be combined throughout the manuscript – Short sentences have been combined throughout the manuscript.
No proper discussion for many sections, for example sections like 3.9, 3.10, 3.11 etc. – We supplemented with adequate discussion or expanded sections.
References should be revised according to journal format. Many scientific names should be in Italics. – We improved to the correct format references and the scientific names were changed to Italics.
Other suggestions are attached (PDF) – Thank You for attached suggestions we provided all them in our MS.

Reviewer 2 Report
The review article foods-1427471 is a well-written and organized research material that addresses the potential health benefits of mushrooms and their contribution in the ethnomedicine. The manuscript has many information on the bio-protective role of C. millitaris and uses evidenced-based literature to make conlusions about its use as food or drug. I have herby some suggestions for authors to improve further their study. These follow the text sequence:
-Graphical abstract
Graphical abstract is mandatory for Foods journal.
-The use of ''they'' throughout the text must be limited and changed to ''these'', given that the authors do not refer to persons.
-Line 442. Change ''abovementioned'' to ''aforementioned''.
-Lines 490, 514, 516, 597, 620, 625, 641, 644, 667, 670, 736, and elsewhere. The citation of the relevant reference in text is not in accordance with Foods guidelines.
Based on the aforementioned, I suggest a minor revision of the review article.
Author Response
Dear Editor,
Thank you very much for reviewing our manuscript. We also thank the Reviewers for carefully reading our manuscript and providing useful comments, which were a great help to us. Based on the Reviewers’ comments, we have revised the manuscript. Bellow our response for Reviewer 2 suggestions:
Reviewer 2
Dear Reviewer thank you for carefully reading our manuscript and providing useful comments, which were a great help to us to improve the MS. Based on yours comments, we have revised the manuscript. Bellow our response for Your suggestions:
The review article foods-1427471 is a well-written and organized research material that addresses the potential health benefits of mushrooms and their contribution in the ethnomedicine. The manuscript has many information on the bio-protective role of C. millitaris and uses evidenced-based literature to make conlusions about its use as food or drug. I have herby some suggestions for authors to improve further their study. These follow the text sequence:
-Graphical abstract
Graphical abstract is mandatory for Foods journal. – We prepared and sent graphical abstract via journal system in our submissions.
-The use of ''they'' throughout the text must be limited and changed to ''these'', given that the authors do not refer to persons. – We changed ''they'' to ''these”
-Line 442. Change ''abovementioned'' to ''aforementioned''. – thank you, we corrected this word.
-Lines 490, 514, 516, 597, 620, 625, 641, 644, 667, 670, 736, and elsewhere. The citation of the relevant reference in text is not in accordance with Foods guidelines. – thank You for suggestions, according to them we improved MS.

Reviewer 3 Report
The manuscript is a review presenting the phytochemistry and biological activities of Cordyceps militaris. In order to be accepted for publishing it needs several major improvements as follows:
- the review is written for scientists, thus some paragraphs can be erased
Abstract
- please rewrite it focusing on C. militaris and not C. sinensis; this is valid throughout the document
- Introduction
- it is too long, some paragraphs repeat the same idea; please condense it
- here is the only place where a comparison between C. militaris and C. sinensis should be made mentioning why C. militaris could succesfully replace C. sinensis
- eliminate Table 1
- what was the methodology of your research?
- Bioactive compounds from militaris
- combine Table 1 and Table 2 (eliminate C. sinensis), include it here
- update with new references (e.g., DOI: 10.1016/j.ejmech.2020.113142)
- Lines 204-221 - erase paragraphs
- Lines 233-238 - erase paragraph
- Lines 249-256 - erase paragraph
- Lines 287-301 - erase paragraph
- Lines 305-323 - condensate the two paragraphs; were they found in this matrix?
- Lines 362-370 - erase paragraph
- Biological activity
- please include one/two tables with the in vitro/in vivo or clinical experiments
- Lines 397-410 - erase paragraphs
- Lines 420-427 - erase paragraph
3.1. More references can be included (e.g. DOI: 10.1155/2015/174616)
3.2. Immunostimulating activity
- other references can be included (see DOI: 10.3389/fphar.2020.575704)
- DOI: 10.1016/j.carbpol.2010.01.017
- DOI: 10.1016/j.intimp.2011.04.001
- DOI: 10.1016/j.carbpol.2014.11.059
3.3. Update/more references can be included:
- PMID: 15132834
- DOI: 10.1016/j.jep.2010.07.020
- DOI: 10.4014/jmb.0800.272
- DOI: 10.1016/j.fct.2010.04.036
- DOI: 10.1155/2017/8474703
- DOI: 10.3390/biom9090407
- DOI: 10.1177/1534735420923756
3.4. Antioxidant activity - Update/more references can be included:
- DOI: 10.1016/j.carbpol.2008.07.023
- DOI: 10.5897/AJMR11.548
- DOI: 10.1016/j.ijbiomac.2013.03.041
- DOI: 10.1271/bbb.100262
- DOI: 10.1021/jf505915t
- Line 529 - “In vitro” instead of “In vivo”
3.5. Anti-inflammatory activity - more references can be included:
- DOI: 10.1016/j.jep.2004.10.009
- DOI: 10.1016/j.ejphar.2006.06.047
3.6. Hypolipidemic activity - more references can be included:
- DOI: 10.1254/jphs.08308fp
- PMID: 21882527
3.7. Hypoglycemic Activity - more references can be included:
- DOI: 10.1016/j.nutres.2015.04.011
- DOI: 10.1371/journal.pone.0166342
- DOI: 10.4162/nrp.2017.11.3.180
- DOI: 10.1016/j.ijbiomac.2020.10.207
Conclusion
- why is Pentostatin mentioned here? It has any connection with the topic?
- this section can be made more powerful
Author Response
Dear Editor,
Thank you very much for reviewing our manuscript. We also thank the Reviewers for carefully reading our manuscript and providing useful comments, which were a great help to us. Based on the Reviewers’ comments, we have revised the manuscript. Bellow our response for Reviewer 3 suggestions:
Reviewer 3
The manuscript is a review presenting the phytochemistry and biological activities of Cordyceps militaris. In order to be accepted for publishing it needs several major improvements as follows:
- the review is written for scientists, thus some paragraphs can be erased
Dear Reviewer thank you for carefully reading our manuscript and providing useful comments, which were a great help to us to improve the MS. Based on yours comments, we have revised the manuscript. Bellow our response for Your suggestions:
Abstract
- please rewrite it focusing on C. militaris and not C. sinensis; this is valid throughout the document – We rewrite abstract according to recommendation.
Introduction
- it is too long, some paragraphs repeat the same idea; please condense it – We condensed paragraphs.
- here is the only place where a comparison between C. militaris and C. sinensis should be made mentioning why C. militaris could succesfully replace C. sinensis
- eliminate Table 1 – We eliminated Table 1.
- what was the methodology of your research? We added methodology.
Bioactive compounds from militaris
- combine Table 1 and Table 2 (eliminate C. sinensis), include it here – We condensed Table 1 and Table 2.
Thank You for suggestions about: erase paragraphs and update references – Almost we erased and added ( also for paragraph 31 – 3.7) to MS:
- Lines 204-221 - erase paragraphs
- update with new references (e.g., DOI: 10.1016/j.ejmech.2020.113142)
- Lines 233-238 - erase paragraph
- Lines 249-256 - erase paragraph
- Lines 287-301 - erase paragraph
- Lines 305-323 - condensate the two paragraphs; were they found in this matrix?
- Lines 362-370 - erase paragraph
Biological activity
- please include one/two tables with the in vitro/in vivo or clinical experiments – We improved MS in paragraphs about in vitro/in vivo research, so the table will be a repetition of information.
- Lines 397-410 - erase paragraphs
- Lines 420-427 - erase paragraph
3.1. More references can be included (e.g. DOI: 10.1155/2015/174616)
3.2. Immunostimulating activity
- other references can be included (see DOI: 10.3389/fphar.2020.575704)
- DOI: 10.1016/j.carbpol.2010.01.017
- DOI: 10.1016/j.intimp.2011.04.001
- DOI: 10.1016/j.carbpol.2014.11.059
3.3. Update/more references can be included:
- PMID: 15132834
- DOI: 10.1016/j.jep.2010.07.020
- DOI: 10.4014/jmb.0800.272
- DOI: 10.1016/j.fct.2010.04.036
- DOI: 10.1155/2017/8474703
- DOI: 10.3390/biom9090407
- DOI: 10.1177/1534735420923756
3.4. Antioxidant activity - Update/more references can be included:
- DOI: 10.1016/j.carbpol.2008.07.023
- DOI: 10.5897/AJMR11.548
- DOI: 10.1016/j.ijbiomac.2013.03.041
- DOI: 10.1271/bbb.100262
- DOI: 10.1021/jf505915t
- Line 529 - “In vitro” instead of “In vivo” – We corrected MS.
3.5. Anti-inflammatory activity - more references can be included:
- DOI: 10.1016/j.jep.2004.10.009
- DOI: 10.1016/j.ejphar.2006.06.047
3.6. Hypolipidemic activity - more references can be included:
- DOI: 10.1254/jphs.08308fp
- PMID: 21882527
3.7. Hypoglycemic Activity - more references can be included:
- DOI: 10.1016/j.nutres.2015.04.011
- DOI: 10.1371/journal.pone.0166342
- DOI: 10.4162/nrp.2017.11.3.180
- DOI: 10.1016/j.ijbiomac.2020.10.207 – Dear Reviewer thank you for suggestions as above: almost of references we added (for paragraph 3.1 – 3.7) to MS.
Conclusion
- why is Pentostatin mentioned here? It has any connection with the topic? – We omitted Pentostatin from this section, thank you for this suggestions.
- this section can be made more powerful – Thank you, We made corrections.

Round 2
Reviewer 1 Report
Authors thoroughly responded to the suggestions made by reviewers and the quality of the manuscript is now improved. Thus, I recommend the present version of the manuscript for publication consideration.
Reviewer 3 Report
The authors thoroughly addressed the questions and suggestions.
I recommend accepting the manuscript in the present form.